# SafePaint: Anti-forensic Image Inpainting with Domain Adaptation

## ABSTRACT

Existing image inpainting methods have achieved remarkable accomplishments in generating visually appealing results, often accompanied by a trend toward creating more intricate structural textures. However, while these models excel at creating more realistic image content, they often leave noticeable traces of tampering, posing a significant threat to security. In this work, we take the anti-forensic capabilities into consideration, firstly proposing an end-to-end training framework for anti-forensic image inpainting named SafePaint. Specifically, we innovatively formulated image inpainting as two major tasks: semantically plausible content completion and region-wise optimization. The former is similar to current inpainting methods that aim to restore the missing regions of corrupted images. The latter, through domain adaptation, endeavors to reconcile the discrepancies between the inpainted region and the unaltered area to achieve anti-forensic goals. Through comprehensive theoretical analysis, we validate the effectiveness of domain adaptation for anti-forensic performance. Furthermore, we meticulously crafted a region-wise separated attention (RWSA) module, which not only aligns with our objective of anti-forensics but also enhances the performance of the model. Extensive qualitative and quantitative evaluations show our approach achieves comparable results to existing image inpainting methods while offering anti-forensic capabilities not available in other methods.

## CCS CONCEPTS

• **Security and privacy → Privacy protections**.

## KEYWORDS

image inpainting, anti-forensics, domain adaptation

## 1 INTRODUCTION

The purpose of image inpainting is to restore the missing regions of damaged images, enabling them to have semantically plausible content. Recently, image inpainting techniques have matured significantly, achieving results that are often imperceptible to the human eye. However, while inspiring, they still exhibit shortcomings. On the one hand, image inpainting techniques have positive applications, such as removing private objects from images [7]. If privacy information in images is exposed by malicious individuals through forensic methods and reverse engineering, it could be disastrous for

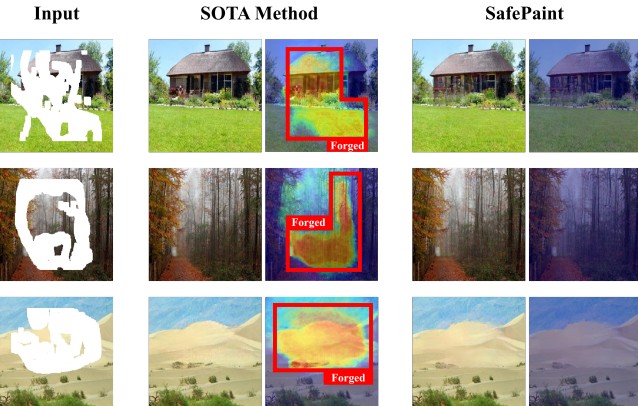

**Figure 1: Examples of images inpainted by SOTA method FcF [19] and our SafePaint, which are selected from the Places2 dataset. Each result of the methods includes an inpainted image and the corresponding heatmap on inpainting detector IID-Net [33].**

stakeholders. On the other hand, as mentioned in [41], commonly used image evaluation metrics in this field (such as PSNR, SSIM, *etc.*) do not adequately represent the performance of image inpainting. An approach that performs better according to these metrics may not necessarily be visually superior to another. The ability of images to resist forensic analysis to some extent aligns with human perceptual judgment and can serve as a basis for measuring the quality of image inpainting. Therefore, the exploration of anti-forensic image inpainting techniques is of considerable significance.

The coarse-to-fine framework [31, 35, 36, 38, 39] is a classic strategy in the field of image inpainting. It resembles human deductive reasoning, gradually restoring the image from easy to difficult, from the outer to the inner, and from superficial to deep. However, in the design of this coarse-to-fine inpainting manner, the functionalities of different networks are approximately similar, which does not fully exploit the advantages. Another research focus in image inpainting is the parallel inpainting of structure and texture [13, 19], allowing models to generate finer structural textures and make the inpainted content more realistic. However, texture structures inconsistent with the background content will attract the attention of forensic detectors, thereby posing a serious threat to the security of the inpainted image.

To further improve the performance of image inpainting methods, many approaches have incorporated attention modules into the models. Traditional spatial attention modules can enhance the network's receptive field and extraction of long-range information. However, for specific tasks such as image inpainting, their performance improvement is often limited. Yu *et al.* drew inspiration from

patch-based image inpainting methods and proposed a context attention module [35, 36] tailored to the characteristics of the image inpainting tasks, simultaneously meeting the requirements of enhancing the network's receptive field and leveraging foreground information. However, its core idea of copy-move is precisely one of the key elements that forensic methods pay attention to, severely restricting the anti-forensic capabilities of the inpainting methods.

In this paper, we propose a new image inpainting architecture to achieve anti-forensic image inpainting tasks. Unlike the previous coarse-to-fine image inpainting methods that simply stack networks for multiple inpainting processes, we adopt a task-decoupled inpainting mode, using an "inpainting first, adjust later" strategy. After completing the inpainting process, we enhance the semantic alignment between the image foreground and background using the domain adaptation method, thereby improving its anti-forensic performance. Furthermore, we carefully designed a region-wise separated attention (RWSA) module, which not only fits our tasks well but also improves the performance of the model. As shown in Fig.1, compared to the state-of-the-art (SOTA) method, our approach possesses strong anti-forensic capabilities.

The main contributions are summarized as follows:

- To our best knowledge, we are the first to introduce an end-to-end training framework for anti-forensic image inpainting, making image inpainting more secure and reliable. Moreover, we innovatively incorporate the anti-forensic capabilities of image inpainting methods as one of the criteria for evaluating the quality of image inpainting, thus compensating for the shortcomings of traditional image inpainting evaluation metrics.
- We have improved the traditional coarse-to-fine image inpainting approaches by fully decoupling the image inpainting and refinement processes. This not only fully utilizes the prior knowledge obtained during the inpainting process, but also significantly enhances the anti-forensic performance of the inpainted images through domain adaptation.
- To meet the requirements of anti-forensic image inpainting tasks and fully exploit the potential of the model, we specifically designed the region-wise separated attention (RWSA) module. This module can also be inserted into other image inpainting models to improve their anti-forensic performance.
- Comprehensive experiments on three widely-used datasets demonstrate that our proposed method significantly outperforms state-of-the-art (SOTA) methods in terms of anti-forensic capabilities, while achieving comparable performance with them in traditional evaluation metrics.

## 2 RELATED WORK

### 2.1 Traditional Inpainting Methods

Traditionally, inpainting methods have two main categories, *i.e.*, diffusion-based and patch-based.

**Diffusion-Based methods.** Diffusion-based methods [1, 2, 5] refer to the utilization of mathematical or physical partial differential equations (PDEs) to smoothly propagate known pixel values from the image's existing regions into the missing areas to restore damaged images.

**Patch-Based methods.** Patch-based image inpainting methods [4, 6–9, 16] involve calculating and searching for samples within the known regions of the damaged images that bear the highest similarity to the missing areas, thereby utilizing them as materials to restore the damaged image. These methods exhibited strong performance in early image inpainting tasks.

However, due to significant time costs and limitations imposed by the limited information of a single image, which makes it challenging to reconstruct semantically rich images, these approaches have gradually been abandoned.

### 2.2 Deep-Learning-Based Inpainting Methods

From the standpoint of design strategy, existing deep-learning-based image inpainting methods can be roughly divided into two types: one-stage and coarse-to-fine methods.

**One-Stage Methods.** Pathak *et al.* [30] were the first to apply the generative adversarial network (GAN) [11] to image inpainting, laying the foundation for using deep learning to tackle image inpainting problems. Iizuka *et al.* [17] improved the consistency of image completion by incorporating global and local discriminators into the network. To address the limitation of traditional convolutional methods, Liu *et al.* [24] proposed partial convolution, which dynamically updates the mask during convolution computation, achieving favorable results for corrupted images with irregularly shaped masks. Zeng *et al.* [37] enhances the contextual reasoning capabilities of the network through carefully designed AOT blocks. To generate content where structure and texture align more closely, work [13] adequately models both types of information in a coupled manner. Inpainting high-resolution images with large missing areas remains a challenge. Therefore, Suvorov *et al.* [32] introduced a network structure based on Fast Fourier convolution (FFC), which has a larger receptive field in the initial stage of the network and performs well in complex image inpainting tasks. Inspired by this work, Jain *et al.* [19] combined the receptive power of Fast Fourier convolution with a co-modulated coarse-to-fine generator, addressing image inpainting problems by simultaneously considering image structures and textures. Due to insufficient prior knowledge, these methods are sometimes plagued by visible artifacts.

**Coarse-to-Fine Methods.** Attention mechanism enables image inpainting models to leverage long-range feature information. Inspired by patch-based methods, Yu *et al.* [35] proposed contextual attention. This module divides the image into several pixel patches and calculates their cosine similarity to guide the inpainting process. Building upon [35], Yu *et al.* [36] introduced gated convolution to address the shortcomings of partial convolution from [24]. Simultaneously, the authors introduced a sample discriminator SN-PatchGAN to generate high-quality inpainting results and accelerate training. Liu *et al.* [25] extended the attention mechanism to missing pixels. Progressive inpainting strategies have become another major technological innovation in the field of image inpainting. Zhang *et al.* [40] divided the image inpainting process into four phases and used a long short-term memory (LSTM) architecture [14] to control the information flow of the progressive process. Nazeri *et al.* [29] drew inspiration from sketching, first generating a hypothetical edge image of the image to be inpainted as prior knowledge, and then filling in the missing region in the next step. Observing that

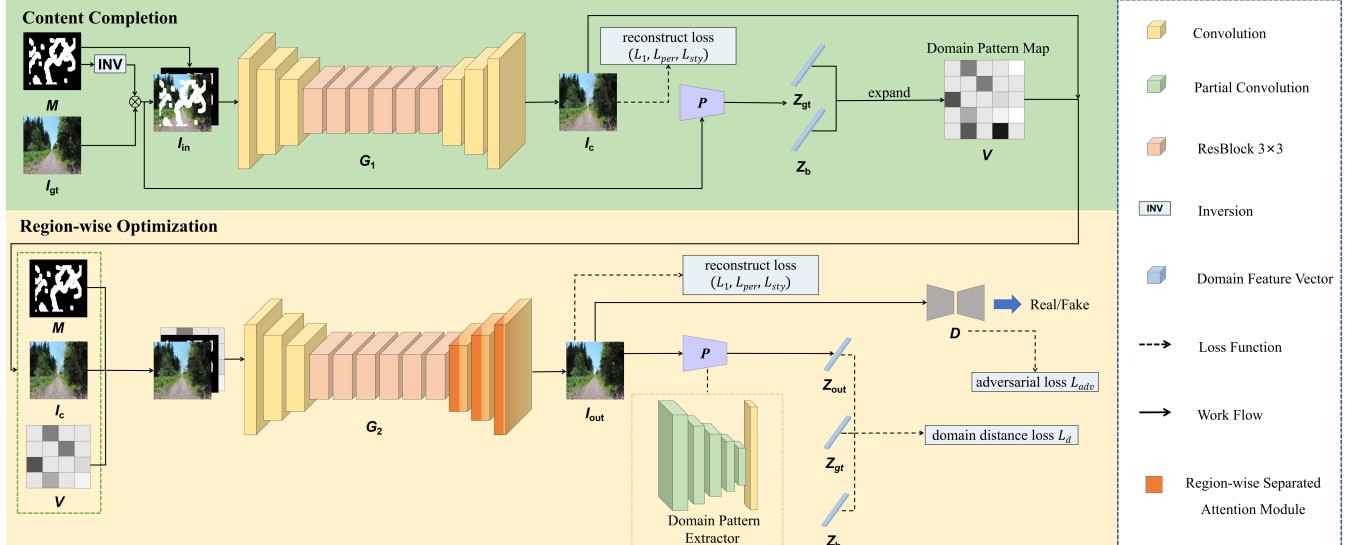

**Figure 2: The overview of our proposed SafePaint. SafePaint adopts an end-to-end architecture that involves two phases. The first stage mainly focuses on content completion, while the second stage is responsible for region-wise optimization, which is the key to enhancing anti-forensic abilities.**

there still exists useful information in the failure cases, Zeng *et al.* [39] proposed an iterative method with feedback mechanisms for inpainting. Reference [44] improved upon the limitations of the specialized convolution in [24, 36], allowing for a more flexible implementation of coarse-to-fine inpainting. Although these methods provide valuable insights into image inpainting, they fail to meet the requirements of anti-forensic image inpainting tasks.

### 2.3 Inpainting Detection Methods

While image inpainting methods are rapidly developing, the advancement of their counterpart, image inpainting detection methods, is also noteworthy. Conventional methods [23, 34] mostly focus on manual features, *e.g.*, the similarities between image patches. Such methods incur high computational cost and often yield minimal effectiveness when confronted with complex scenarios. Recently, Many methods [22, 33] have shifted their focus toward deep-learning-based inpainting techniques. By automatic feature extraction, these methods are capable of achieving precise detection in even more complex situations. Moreover, many methods [12, 26] are proposed to detect complicated combinations of forgeries (including inpainting).

However, these detection methods start from a fundamentally similar point, which is leveraging the inconsistency between the distribution of generated data and that of original data.

### 2.4 Domain Adaptation

The objective of domain adaptation is to achieve the transformation from a source domain to a target domain. The pioneering work Pix2Pix [18] demonstrated remarkable results in image-to-image (I2I) translation tasks by leveraging paired images and conditional generative adversarial networks (GAN). To overcome the reliance on paired image data, Zhu *et al.* introduced CycleGAN [43]. The

aforementioned methods require domain labels; in contrast, the methods [3, 20] necessitate no domain labels. In image inpainting, domain labels are typically not explicitly available. Therefore, it is proposed to use unlabeled domain adaptation aimed at enhancing the anti-forensic capabilities of our method.

## 3 METHOD

### 3.1 Model Architecture

As shown in Fig.2, our SafePaint achieves anti-forensic inpainting through two subtasks, *i.e.*, content completion and region-wise optimization. We will introduce the generators, the discriminator, and the domain pattern extractor of the model below.

**Generator and Discriminator.** We propose an image inpainting network that consists of two stages. It includes a generator in each stage and a discriminator in stage $II$. The two generators $G_1$, $G_2$ follow an architecture similar to the method proposed by Johnson *et al.* [21] that we use only the image generator as our backbone generator. In the upsampling process of $G_2$, we further integrated the RWSA module to facilitate domain adaptation and enhance performance. For the discriminator, We apply the Patch-GAN [18, 43] instead of vanilla GAN to classify each image patch as pristine or inpainted. To improve the stability of the training, we also introduced Spectral normalization (SN) [28] to both the generators and the discriminator.

**Domain Pattern Extractor.** We design the domain pattern extractor to implement domain adaptation. Considering that the background and foreground regions have irregular shapes, we replaced vanilla convolutional layers with partial convolutional layers [24], which are specially designed for image inpainting with irregular masks. The output of the network will be represented in vector form, serving to characterize features from different domains.

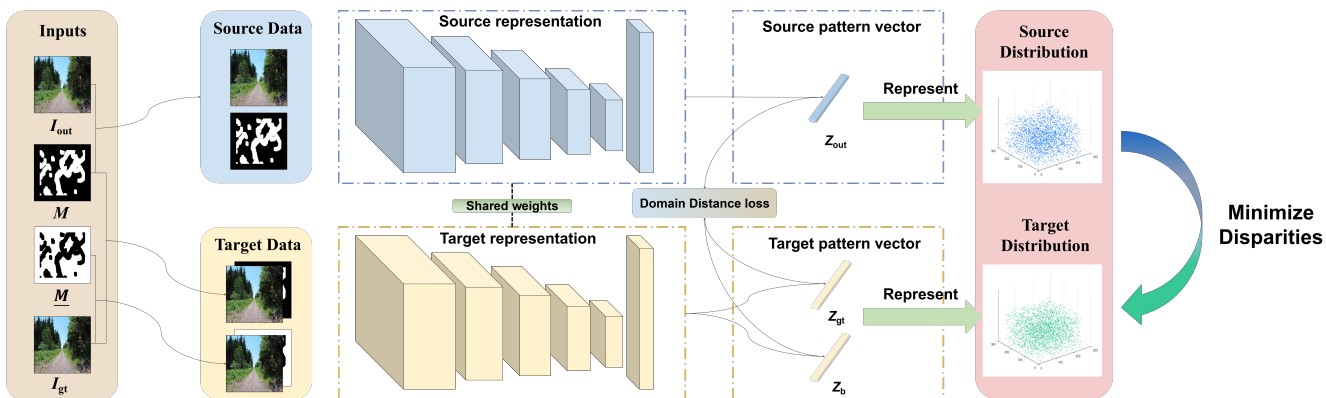

Figure 3: The implemental detail of our domain adaptation.

## 3.2 Domain Adaptation for Image Inpainting

To achieve the goal of anti-forensics, we investigate the commonalities of inpainting methods and the shared properties among their corresponding forensic methods by analyzing massive corrupted image samples. It is drawn that detectors are more inclined to perceive regions where data distribution significantly deviates from that of real image data as tampered areas, a point we will further explore in Section 3.3. Hence, we employ domain adaptation as a means to narrow the gap between the foreground and background of the image, thereby evading the detection of forensic detectors.

We refer to the area to be inpainted as the foreground, and the original area as the background. Specifically, we design a domain pattern extractor to characterize the features within the delineated regions. For image inpainting, the background region is generally not allowed to be modified. In that case, the background region in images to be inpainted is considered as unified across different stages.

As shown in Fig.3, we extracted three different domain pattern vectors $Z_b$, $Z_{gt}$, and $Z_{out}$ by the domain pattern extractor, representing background domain features, foreground domain features of ground-truth, and foreground domain features of the output image, respectively. Since domain pattern vectors can represent the data distributions within specific regions of an image, intuitively, minimizing the disparities between them can be equivalent to reducing the distance between their corresponding domain pattern vectors. To achieve this, we design a special loss function, which will be explained specifically in Section 3.5.

## 3.3 Interpretation for Effectiveness of Domain Adaptation in Anti-forensics

In this section, we will explain how our design of domain adaptation can effectively achieve anti-forensics by analyzing the principles of inpainting methods and their respective forensic approaches. We categorize inpainting methods into diffusion-based, patch-based, and deep-learning-based methods, and we will discuss each of them separately below.

**Diffusion-Based Methods.** Taking the work [5] as an example, it leverages partial differential equations (PDEs) for image inpainting. The core idea is to gradually propagate pixels from the known regions of the image to the areas requiring inpainting. The algorithm can be formulated as follows:

$$I^{n+1}(i, j) = I^n(i, j) + \Delta v I_t^n(i, j), \forall (i, j) \in \Omega \quad (1)$$

where $n$ denotes the number of iterations for the pixel $I(i, j)$ at the coordinate $(i, j)$, $\Delta v$ represents the update rates, $\Omega$ signifies the region to be inpainted, and $I_t^n(i, j)$ is the updated value, calculated by the following formula:

$$I_t^n(i, j) = \overrightarrow{\Delta L^n}(i, j) \cdot \overrightarrow{N}^n(i, j) \quad (2)$$

where $\overrightarrow{\Delta L^n}(i, j)$ measure the change of $L^n(i, j)$, which denotes the propagated information. $\overrightarrow{N}^n(i, j)$ is the propagation direction. Since the diffusion process runs only inside $\Omega$, as the propagation progresses, $I_t^n(i, j)$ will decrease gradually and converge to 0. That means $I(i, j)$ will remain constant in the direction of $\overrightarrow{N}$. However, this phenomenon is extremely rare in the known regions of the image. The forensic method [23] specific to this inpainting mode confirms this, as it locates the inpainting area by comparing the local variances of the inpainted region and the known region.

**Patch-Based Methods.** The design of the work [7] is to find the most suitable patch within the known region to fill the area to be inpainted. It repeats the following three steps: (1) Computing patch priorities. (2) Propagating texture and structure information. (3) Filling region to be inpainted. We define $\Psi_{\hat{f}}$ as the patch with the highest priority, and then we search in the known region for that patch $\Psi_{\hat{b}}$ which is most similar to $\Psi_{\hat{f}}$, calculated by following rule:

$$\Psi_{\hat{b}} = \arg \min_{\Psi_b \in \Phi} d(\Psi_{\hat{f}}, \Psi_b) \quad (3)$$

where $\Omega$ represents the region to be filled, the distance $d(\Psi_m, \Psi_n)$ between two patches $\Psi_m$ and $\Psi_n$ is simply computed using the Euclidean distance. This rigid computational approach may introduce abnormal similarity between the inpainting region and the rest of the image, as the patch block set of the inpainting region may exhibit a one-to-many or many-to-many relationship with that of the known region. Wu *et al.* [34] utilizes this characteristic by identifying sets of block pairs exhibiting abnormal similarity as a basis for locating the area that has been inpainted.

**Deep-Learning-Based Methods.** While deep-learning-based inpainting methods mainly rely on data-driven approaches, their basic principles predominantly derive from traditional inpainting methods. For example, the contextual attention mechanism used in [35, 36] draws inspiration from patch-based inpainting methods, while the search and generate process in [25] embodies the properties of diffusion-based inpainting methods. Although these inpainting methods differ, their core ideas are nearly identical, namely, making full use of existing pixel information in images. On the other side of the coin, their corresponding forensic methods also share similarities, primarily utilizing inconsistencies of the statistical distributions between real data and generated data.

Based on the analysis above, we eventually validate the effectiveness of the proposed domain adaptation for image inpainting. Let $X_\Omega$ the set of pixel values in the inpainting area, and $X_\Phi$ denote the set of pixel values in the known area. Then we use $P(X_\Omega)$ and $P(X_\Phi)$ to represent their respective probability distribution. The Kullback-Leibler divergence (KL divergence) is utilized to express the difference between them, formulated as:

$$D_{\mathrm{KL}}(P(X_\Phi)\|P(X_\Omega)) = \int_{-\infty}^{\infty} P(X_\Phi(x)) \log\left(\frac{P(X_\Phi(x))}{P(X_\Omega(x))}\right) dx \quad (4)$$

We denote the domain adaptation transformation as function $T$. Our objective is to find a $T$ that minimizes the KL divergence between these two distributions, thereby achieving the purpose of anti-forensics.

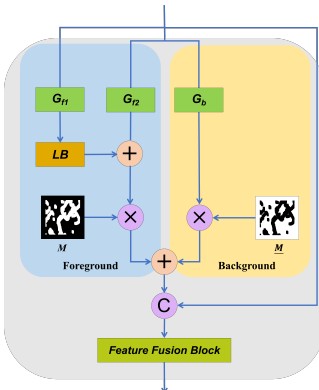

**Figure 4: The detailed structure of our RWSA module.**

## 3.4 Region-wise Separated Attention Module

As illustrated in Fig.4, given an intermediate feature map $x \in \mathbb{R}^{C \times H \times W}$, and the binary mask $M \in \mathbb{R}^{1 \times H \times W}$ of the inpainted region as the inputs (C is the feature channel, H and W is the height and width of the feature map respectively), three 1D channel attention modules $G_b, G_{f_1}, G_{f_2}$ are utilized to weight the input features for different purposes, with the help of foreground region mask $M$ and background region mask $M'$. In detail, $G_b$ aims to select the features that are necessary for image rebuild in the background region. $G_{f_1}$ intends to weigh the input features that have differences between the input and target. Moreover, we design the third channel attention module $G_{f_2}$ to weight features that are unnecessary

to change in the foreground region. Additionally, for the output features of $G_{f_1}$, we add a learnable block $LB$ to learn the domain adaptation. Therefore, the output features of foreground $\mathscr{F}_f$ and background $\mathscr{F}_b$ can be respectively formalized as:

$$\mathscr{F}_f = M \times [LB(G_{f_1}(x)) + G_{f_2}(x)]$$

$$\mathscr{F}_b = (1 - M) \times G_b(x) \quad (5)$$

The core component of proposed RWSA module are channel attention modules ($G_b, G_{f_1}, G_{f_2}$), learnable block $LB$, and Feature Fusion Block $FFB$. Regarding the three channel attention modules, we utilize a modified Squeeze-and-Excitation Network [15] in which we add a MaxPooling layer. The learnable block $LB$ consists of 3 × 3 convolutional layers with two CONV-BN-ELU blocks to learn the textural and structural changes in the inpainted region. In order to fully integrate the feature obtained, we have designed the feature fusion block $FFB$, which concatenates the sum of $\mathscr{F}_f$ and $\mathscr{F}_b$ with the input $x$ in channel axis. Subsequently, the channel size will halve to complete the fusion operation. Above all, our region-wise separated attention (RWSA) module between input features $x$ and output features $\mathscr{F}$ can be finally formalized as:

$$\mathscr{F} = FFB(Concat(\mathscr{F}_f + \mathscr{F}_b, x)) \quad (6)$$

where $Concat$ is the implementation of concatenation.

## 3.5 Loss Functions

The model training contains the $\ell_1$ loss, perceptual loss, style loss, adversarial loss, and domain distance loss, to achieve visual plausibility and anti-forensic performance.

Formally, we designate the two generators as $G_1$ and $G_2$ respectively and denote the discriminator in stage $II$ as $D$. Let $I_{gt}$ be the ground truth image, $M$ be the input binary mask which represents the region to be inpainted, $I_{in} = I_{gt} \odot (1 - M) + M$ be the input image, $V$ be the domain pattern map. Then the coarse result $I_c$ and final output $I_{out}$ are formulated as:

$$I_c = I_{gt} \odot (1 - M) + G_1(Concat(I_{in}, M)) \odot M$$

$$I_{out} = I_{gt} \odot (1 - M) + G_2(Concat(I_c, M, V)) \odot M \quad (7)$$

$\ell_1$ **Loss.** We adopt the $\ell_1$ loss to calculate difference between $I_c$ and $I_{gt}$ as well as the difference from $I_{out}$ to $I_{gt}$, formulated as:

$$\mathscr{L}_1 = \mathbb{E}\left[\|I_c - I_{gt}\|_1 + \|I_{out} - I_{gt}\|_1\right] \quad (8)$$

**Perceptual Loss.** For image inpainting, the perceptual loss from a pre-trained and fixed VGG-19 is used. The perceptual loss compares the difference between the deep feature map of the generated image and the ground truth. Such a loss function can compensate for the weakness of $\ell_1$ Loss that helps the model capture high-level semantics. The perceptual loss is defined as:

$$\mathscr{L}_{per} = \mathbb{E}\left[\sum_i \|\phi_i(I_c) - \phi_i(I_{gt})\|_1 + \|\phi_i(I_{out}) - \phi_i(I_{gt})\|_1\right] \quad (9)$$

**Style Loss.** For the purpose of guaranteeing style coherence, we use style loss to compute the distance between feature maps. The style loss can be illustrated as:

$$\mathscr{L}_{sty} = \mathbb{E}\left[\sum_i \|\psi_i(I_c) - \psi_i(I_{gt})\|_1 + \|\psi_i(I_{out}) - \psi_i(I_{gt})\|_1\right] \quad (10)$$

Anonymous Authors

where $\psi_i(\cdot) = \phi_i(\cdot)^T \phi_i(\cdot)$ denotes the Gram matrix constructed from the activation map $\phi_i$.

**Adversarial Loss.** Adversarial Loss is designed to help the model generate plausible images. Given the characteristic of our task, the discriminator $D$ takes $I_{gt}$ as real images and $I_{out}$ as fake images. The adversarial loss is as follows:

$$\mathscr{L}_{adv} = \mathbb{E}[\max(0, 1 - D(I_{gt}))] + \mathbb{E}[\max(0, 1 + D(I_{out}))] \quad (11)$$

**Domain Distance Loss.** Enhancing the alignment between the foreground and background is a crucial step in our SafePaint, and we accomplish this goal by incorporating the domain distance loss. Firstly, the character $P$ is used to denote the domain pattern extractor. As mentioned in section 3.2, the three different domain pattern vectors $Z_b, Z_{gt}, Z_{out}$ can be computed by:

$$Z_b = P(I_{gt}, \overline{M}), \quad Z_{gt} = P(I_{gt}, M), \quad Z_{out} = P(I_{out}, M) \quad (12)$$

where $M$ represents the input binary mask and $\overline{M}$ represents its reverse form.

After inpainting, we attempt to make the background of the result as close to the foreground as possible while ensuring the quality of inpainted images. Therefore, we bring $Z_{out}$ and $Z_b$ closer while draw $Z_{out}$ and $Z_{gt}$ nearer by following domain distance loss:

$$\mathscr{L}_d = \mathbb{E}[d(Z_{out}, Z_b) + d(Z_{out}, Z_{gt})] \quad (13)$$

in which $d(\cdot, \cdot)$ is Euclidean distance.

To summarize, the total loss can be written correspondingly as:

$$\mathscr{L}_{total} = \lambda_1 \mathscr{L}_1 + \lambda_{per} \mathscr{L}_{per} + \lambda_{sty} \mathscr{L}_{sty} + \lambda_{adv} \mathscr{L}_{adv} + \lambda_d \mathscr{L}_d \quad (14)$$

where $\lambda_1, \lambda_{per}, \lambda_{sty}, \lambda_{adv}, \lambda_d$ are the weights for respective loss terms.

## 4 EXPERIMENTS

### 4.1 Experimental Settings

We evaluate the proposed method on the Places2 [42], Paris-Street-View [10], CelebA [27] datasets, which are commonly used in the field. NVIDIA Irregular Mask [18] is used for training and testing and we classified the mask data based on their mask ratio relative to the image with an increment of 10%. All the images and corresponding masks are resized to 256 × 256. We train our model in pytorch. Training is launched on a single NVIDIA RTX 4090 GPU with the batch size of 4, optimized by the Adam optimizer with $betas = (0.5, 0.999)$. We adjust the learning rate as $2 \times 10^{-4}$ for both discriminator and generator. Besides, $\lambda_1, \lambda_{per}, \lambda_{sty}, \lambda_{adv}, \lambda_d$ in Eq.(14) are set to 1, 0,1, 250, 0.1, 0.01.

### 4.2 Comparisons with State-of-the-art Methods

We compare our SafePaint on the Places2 [42], Paris-Street-View [10], CelebA [27] datasets with the state-of-the-art methods introduced previously: EdgeConnect [29], MADF [44], CTSDG [13], AOT-GAN [37], Lama [32], FcF [19]. We use their official implementation and the models provided by the authors. As traditional image quality assessment metrics, LPIPS and PSNR are used for quantitative quality evaluation. Since this is a piece focusing on anti-forensics, we take the anti-forensic ability of inpainting methods into account. Therefore, we adopt two pre-trained image forgery detectors PSCC-Net [26] and TruFor [12], and one inpainting-specific detector [33]

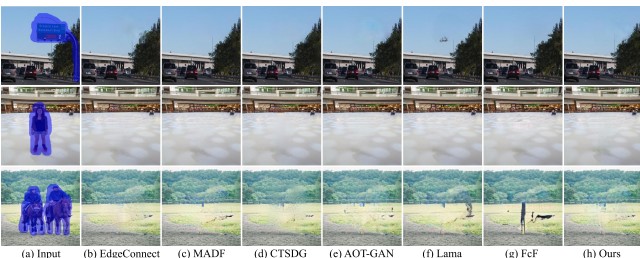

(a) Input  (b) EdgeConnect  (c) MADF  (d) CTSDG  (e) AOT-GAN  (f) Lama  (g) FcF  (h) Ours

**Figure 5: Visual comparison of our SafePaint with EdgeConnect [29], MADF [44], CTSDG [13], AOT-GAN [37], Lama [32], FcF [19] on dataset Places2.**

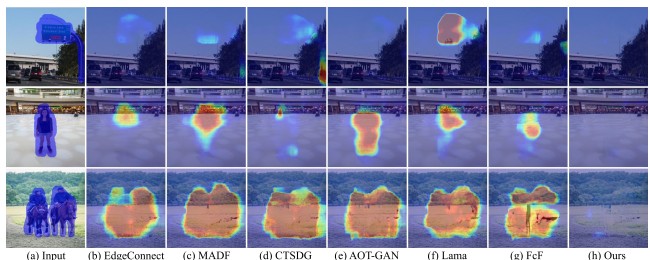

(a) Input  (b) EdgeConnect  (c) MADF  (d) CTSDG  (e) AOT-GAN  (f) Lama  (g) FcF  (h) Ours

**Figure 6: Visualization of anti-forensic capabilities comparisons on detector PSCC-Net [26].**

for anti-forensics analysis. To quantify the anti-forensic performance, we use pixel-level Area Under Curve (AUC), F1 score on manipulation masks, and the accuracy of detectors as main metrics. Since the inpainted images serve as negative samples, lower AUC, F1, and ACC indicate better anti-forensic performance.

**Quantitative evaluation on visual quality.** Table 1 reports the numerical comparisons of proposed SafePaint with EdgeConnect [29], MADF [44], CTSDG [13], AOT-GAN [37], Lama [32], and FcF [19] across the three datasets. The comparisons are based on two image quality metrics: LPIPS and PSNR. It can be observed that although we compromise visual quality to a certain extent to enhance image security in stage $II$, the proposed network is comparable to state-of-the-art methods in visual comparison evaluation, and the inpainted images remain visually appealing.

**Qualitative evaluation on visual quality.** Fig.5 shows the visual quality results of the SafePaint compared with existing methods on the dataset Places2. It can be seen, as one of the representative works of parallel structural and textural image inpainting, FcF [19] exhibits powerful detail generation capabilities, which sometimes paradoxically lead to content generation that is discordant with the surrounding areas. For other methods, the inpainted content may deviate from the known regions to some extent. Owing to our domain adaptation design, our SafePaint can generate more harmonious content.

**Anti-forensics analysis.** Anti-forensic performance signifies the ability of image inpainting methods to evade the detection of forensic detectors. We select two representative image forgery detectors and one inpainting detector to evaluate the anti-forensic ability of our SafePaint and the comparison methods, including

**Table 1: Quantitative comparison on visual quality.**

| Dataset | | Places2 | | | Paris Street View | | | CelebA | | |
|---|---|---|---|---|---|---|---|---|---|---|
| Mask Ratio | | 10%-20% | 30%-40% | 50%-60% | 10%-20% | 30%-40% | 50%-60% | 10%-20% | 30%-40% | 50%-60% |
| PSNR↑ | EdgeConnect [29] | 27.31 | 22.37 | 18.58 | 30.37 | 25.68 | 21.62 | 30.35 | 25.40 | 20.67 |
| | MADF [44] | **29.18** | 23.46 | 19.15 | **32.71** | **27.15** | **22.37** | 30.86 | 25.58 | 20.34 |
| | CTSDG [13] | 29.02 | 23.34 | 19.14 | 32.65 | 26.88 | 22.26 | 30.90 | 25.84 | 21.08 |
| | AOT-GAN [37] | 28.97 | 23.50 | 18.89 | / | / | / | 31.31 | 25.42 | 20.09 |
| | Lama [32] | 29.17 | **23.56** | **19.28** | / | / | / | **32.37** | 26.95 | **22.50** |
| | FcF [19] | 28.63 | 22.71 | 18.19 | / | / | / | 32.30 | **27.01** | 22.48 |
| | SafePaint | 29.00 | 23.37 | 19.03 | 32.45 | 27.12 | 22.20 | 32.06 | 26.76 | 21.63 |
| LPIPS↓ | EdgeConnect [29] | 0.053 | 0.136 | 0.262 | 0.053 | 0.117 | 0.233 | 0.043 | 0.091 | 0.178 |
| | MADF [44] | **0.037** | 0.105 | 0.230 | **0.033** | **0.090** | 0.205 | 0.052 | 0.102 | 0.212 |
| | CTSDG [13] | 0.049 | 0.145 | 0.298 | 0.037 | 0.111 | 0.247 | 0.057 | 0.112 | 0.206 |
| | AOT-GAN [37] | 0.045 | 0.110 | 0.234 | / | / | / | 0.026 | 0.075 | 0.188 |
| | Lama [32] | **0.037** | **0.102** | **0.218** | / | / | / | **0.020** | **0.054** | **0.113** |
| | FcF [19] | 0.039 | 0.109 | 0.231 | / | / | / | 0.027 | 0.058 | 0.114 |
| | SafePaint | 0.041 | 0.116 | 0.246 | 0.038 | 0.092 | **0.200** | 0.032 | 0.068 | 0.138 |

**Table 2: Anti-forensic performance evaluation of our SafePaint with SOTAs on detector PSCC-Net [26].**

| Metrics | AUC↓ | | | F1↓ | | | ACC↓ | | |
|---|---|---|---|---|---|---|---|---|---|
| Mask Ratio | 10%-20% | 30%-40% | 50%-60% | 10%-20% | 30%-40% | 50%-60% | 10%-20% | 30%-40% | 50%-60% |
| EdgeConnect [29] | 0.7376 | 0.7790 | 0.8388 | 0.1056 | 0.1437 | 0.2458 | 0.2038 | 0.3890 | 0.4970 |
| MADF [44] | 0.5738 | 0.8407 | 0.8945 | 0.0664 | 0.4724 | 0.7141 | 0.2272 | 0.7240 | 0.8988 |
| CTSDG [13] | 0.8034 | 0.8412 | 0.8533 | 0.2531 | 0.4080 | 0.4930 | 0.3060 | 0.6836 | 0.7428 |
| AOT-GAN [37] | 0.7607 | 0.8067 | 0.8661 | 0.1784 | 0.2487 | 0.4440 | 0.2724 | 0.5528 | 0.8760 |
| Lama [32] | 0.6800 | 0.7529 | 0.8084 | 0.0592 | 0.1226 | 0.2236 | 0.0790 | 0.2548 | 0.3432 |
| FcF [19] | 0.6490 | 0.7448 | 0.8084 | 0.0344 | 0.1182 | 0.3005 | 0.0598 | 0.5196 | 0.9010 |
| SafePaint | **0.5628** | **0.5729** | **0.6058** | **0.0153** | **0.0092** | **0.0106** | **0.0166** | **0.0110** | **0.0132** |

PSCC-Net [26], TruFor [12] and IID-Net [33]. Table 2, Table 3 and Table 4 illustrate the comparisons regarding their anti-forensic performance on Places2. The AUC, F1, and ACC with smaller values indicate better anti-forensic ability. We can see that our model always outperforms the state-of-the-art counterparts by a significant margin. The visual results are depicted via the heatmap in Fig.6. It is evident that compared to the other methods, our approach leaves very faint tampering traces in the image, showing excellent anti-forensic capabilities.

## 4.3 Ablation Study

In this section, we will investigate the effectiveness of each component in the SafePaint, including domain distance loss (DDL) and region-wise separated attention (RWSA) module proposed by us. Given their design for anti-forensic purposes, we will only compare the detection resistance performance of each model. Specifically, the ablation experiments are conducted on the dataset Places2 and the mask ratio is set to 30%-40%. The results of ablating each component are reported in Table 5, representing their detection evasion

performance on detectors PSCC-Net, TruFor and IID-Net. We denote our SafePaint without DDL and RWSA module as "Backbone". It can be observed that "Backbone+DDL" outperforms "Backbone", demonstrating the benefit of using domain distance loss. Another observation is that "Backbone+RWSA" performs more favorably than "Backbone", which indicates the advantage of the region-wise separated attention (RWSA) module. Finally, our full method SafePaint, *i.e.*, "Backbone+DDL+RWSA", achieves the best results.

## 4.4 Generality of RWSA

Based on considerations regarding anti-forensics and the requirements stemming from domain adaptation, we meticulously design the RWSA module. It is natural to elegantly insert the RWSA in any existing networks to enhance their anti-forensic performance. To substantiate this assertion, we will quantitatively compare the original versions and updated versions of some state-of-the-art methods, while keeping the experimental setup consistent with Section 4.3. To be precise, we conduct experiments on EdgeConnect [29] and AOT-GAN [37]. The training strategy also adopts their original

**Table 3: Anti-forensic performance evaluation of our SafePaint with SOTAs on detector Trufor [12].**

| Metrics | AUC↓ | | | F1↓ | | | ACC↓ | | |
|---|---|---|---|---|---|---|---|---|---|
| Mask Ratio | 10%-20% | 30%-40% | 50%-60% | 10%-20% | 30%-40% | 50%-60% | 10%-20% | 30%-40% | 50%-60% |
| EdgeConnect [29] | 0.8737 | 0.8275 | 0.8221 | 0.4432 | 0.3627 | 0.2208 | 0.6770 | 0.4304 | 0.2088 |
| MADF [44] | 0.8557 | 0.8151 | 0.8185 | 0.4175 | 0.3720 | 0.3226 | 0.6972 | 0.5200 | 0.3736 |
| CTSDG [13] | 0.8599 | 0.8249 | 0.8229 | 0.4351 | 0.4211 | 0.3340 | 0.7412 | 0.5846 | 0.3244 |
| AOT-GAN [37] | 0.8650 | 0.8232 | 0.8294 | 0.4327 | 0.3923 | 0.3012 | 0.7456 | 0.5622 | 0.3294 |
| Lama [32] | 0.8610 | 0.8216 | 0.8195 | 0.4303 | 0.4007 | 0.3528 | 0.7402 | 0.6238 | 0.4866 |
| FcF [19] | 0.8577 | 0.8172 | 0.8101 | 0.4078 | 0.3400 | 0.2314 | 0.6656 | 0.4734 | 0.2766 |
| SafePaint | **0.6417** | **0.5781** | **0.5771** | **0.0983** | **0.0624** | **0.0620** | **0.1934** | **0.1290** | **0.1488** |

**Table 4: Anti-forensic performance evaluation of our SafePaint with SOTAs on detector IID-Net [33].**

| Metrics | AUC↓ | | | F1↓ | | | ACC↓ | | |
|---|---|---|---|---|---|---|---|---|---|
| Mask Ratio | 10%-20% | 30%-40% | 50%-60% | 10%-20% | 30%-40% | 50%-60% | 10%-20% | 30%-40% | 50%-60% |
| EdgeConnect [29] | 0.7506 | 0.7845 | 0.7926 | 0.1143 | 0.1582 | 0.1743 | 0.3701 | 0.5090 | 0.4928 |
| MADF [44] | 0.7014 | 0.7234 | 0.7465 | 0.0735 | 0.0837 | 0.1149 | 0.1948 | 0.2875 | 0.3652 |
| CTSDG [13] | 0.7437 | 0.8059 | 0.8288 | 0.0921 | 0.2433 | 0.2516 | 0.3354 | 0.5978 | 0.7315 |
| AOT-GAN [37] | 0.8342 | 0.8617 | 0.8670 | 0.2709 | 0.3156 | 0.3218 | 0.7560 | 0.7913 | 0.8081 |
| Lama [32] | 0.7655 | 0.7801 | 0.7954 | 0.0836 | 0.1078 | 0.1257 | 0.3652 | 0.5484 | 0.6589 |
| FcF [19] | 0.7928 | 0.8254 | 0.8319 | 0.1624 | 0.2011 | 0.2092 | 0.6136 | 0.7032 | 0.7423 |
| SafePaint | **0.5909** | **0.6181** | **0.6195** | **0.0398** | **0.0699** | **0.0722** | **0.0255** | **0.0283** | **0.0319** |

**Table 5: Ablation study conducted on three detectors.**

| Detectors | Model | AUC↓ | F1↓ | ACC↓ |
|---|---|---|---|---|
| PSCC-Net | Backbone | 0.5810 | 0.0395 | 0.5054 |
| | Backbone+DDL | 0.5753 | 0.0119 | 0.0137 |
| | Backbone+RWSA | 0.5778 | 0.0153 | 0.0182 |
| | Backbone+DDL+RWSA | **0.5729** | **0.0092** | **0.0110** |
| Trufor | Backbone | 0.6223 | 0.0659 | 0.6800 |
| | Backbone+DDL | 0.5907 | 0.0653 | 0.2310 |
| | Backbone+RWSA | 0.6064 | 0.0641 | 0.2114 |
| | Backbone+DDL+RWSA | **0.5781** | **0.0624** | **0.1290** |
| IID-Net | Backbone | 0.6596 | 0.0933 | 0.5785 |
| | Backbone+DDL | 0.6339 | 0.0791 | 0.0964 |
| | Backbone+RWSA | 0.6420 | 0.0814 | 0.1297 |
| | Backbone+DDL+RWSA | **0.6181** | **0.0699** | **0.0283** |

**Table 6: Anti-forensic performance evaluation of existing methods and these methods with our RWSA module conducted on three detectors.**

| Detectors | Model | AUC↓ | F1↓ | ACC↓ |
|---|---|---|---|---|
| PSCC-Net | EdgeConnect [29] | 0.7790 | 0.1437 | 0.3890 |
| | EdgeConnect+RWSA | 0.6192 | 0.0666 | 0.1706 |
| | AOT-GAN [37] | 0.8067 | 0.2487 | 0.5528 |
| | AOT-GAN+RWSA | 0.7590 | 0.1110 | 0.2338 |
| Trufor | EdgeConnect | 0.8275 | 0.3627 | 0.4304 |
| | EdgeConnect+RWSA | 0.7066 | 0.1230 | 0.1552 |
| | AOT-GAN | 0.8232 | 0.3923 | 0.5622 |
| | AOT-GAN+RWSA | 0.8159 | 0.3847 | 0.5478 |
| IID-Net | EdgeConnect | 0.7845 | 0.1582 | 0.5090 |
| | EdgeConnect+RWSA | 0.7621 | 0.1387 | 0.4389 |
| | AOT-GAN | 0.8617 | 0.3156 | 0.7913 |
| | AOT-GAN+RWSA | 0.8206 | 0.2713 | 0.7162 |

setting. In line with our approach, we incorporate RWSA into the downsampling layer of the inpainting network. The corresponding numerical results are shown in Table 6. It is evident that RWSA can serve as an effective supplement to address insufficient resistance of traditional inpainting methods against detection.

# 5 CONCLUSIONS

In this work, we propose the first end-to-end learning architecture called SafePaint for anti-forensic inpainting, which is achieved with the help of domain adaptation. The task-decoupled strategy ensures that each stage of the network performs its designated role, while domain adaptation achieves compatibility between the background and the foreground of an image. Moreover, the region-wise separated attention module effectively addresses the shortcomings of the previous attention mechanism module and can be naturally integrated as a plug-in into existing inpainting methods to improve their anti-forensic capabilities. Extensive experiments have demonstrated that our SafePaint has very strong anti-forensic ability while maintaining commendable inpainting performance.

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
