# OpenReview forum: "SafePaint: Anti-forensic Image Inpainting with Domain Adaptation"
_acmmm.org/ACMMM/2024/Conference — MM2024 Oral_

### Official Review · Reviewer_HqtJ · 2024-05-20

**Rating:** 4
**Confidence:** 3

**Summary:**

In this paper, the authors explore a new inpainting scheme from the perspective of creating realistic image content and leaving no traces of tampering. Briefly, it first performs inpainting to recover the missing regions of a corrupted image. Subsequently, the difference between internal and unaltered regions is reconciled by domain adaptation to avoid leaving traces. Experimental results demonstrate its high image quality while avoiding detection.

**Strengths:**

The authors' proposal to use the anti-forensic capability as one of the criteria for evaluating the quality of image inpainting is valuable.
This article is well written, easy to understand and sufficiently experimental.

**Limitations:**

1. The background research is not extensive enough. In the current research, diffusion model for inpainting is effective [1,2], but this paper does not mention the work in this area. So, how does the diffusion model perform for inpainting in the context of anti-forensics? What is the result of this paper comparing with diffusion model in terms of image quality and anti-forensics?
2. Is end-to-end training inpainting necessary and how generalizable is it in the context of diffusion models with strong prior knowledge of image?
3. Lacking an original design, this paper is more akin to a combination of two works. The authors claim to be task-decoupled, however, as shown by the insufficient results in Table 1, the separate tasks seem to be imperfect. There seems to be a loss of image quality in order to achieve anti-forensics.
4. As an architecture for adversarial generative networks, common sense dictates that an adversarial training form of the game should be used, whereas this paper seems to use supervised learning (as in Equation 14). The authors are requested to clarify.
5. Does the domain pattern extractor need to be trained? What are the experimental details?

 Minor issue
1. For easy comparison, authors may underline the suboptimal experimental results.

[1] 10.1109/CVPR52688.2022.01117
[2] 10.1109/PCEMS58491.2023.10136091

**Suitability:**

2

---

### Official Review · Reviewer_6Z84 · 2024-05-24

**Rating:** 5
**Confidence:** 3

**Summary:**

The paper targets the problem that existing image inpainting methods fail to resist forensic analysis, leading to security vulnerabilities. The theoretical proof reveals that the root cause of poor anti-forensic capability for an image lies in the inconsistency of statistical distributions between the inpainted and original parts. Based on this foundation, the authors introduce the domain adaptation method into image inpainting to achieve the goal of anti-forensics. There are two kinds of experiments, i.e., evaluation on visual quality and anti-forensic analysis, and the results prove the effectiveness of the proposed framework.

**Strengths:**

- The topic holds significant importance in the community from the perspective of security.
- The paper makes a strong contribution and has great guiding significance for the follow-up work.
- This paper explores an interesting and exciting problem in image inpainting, pioneeringly taking the anti-forensic capabilities into consideration.
- Comprehensive theoretical explanation, extensive experimental validation.

**Limitations:**

- Some details are not adequately explained (e.g., the criteria for defining metric ACC).
- The adversarial attack method can also deceive the detector to achieve the goal of anti-forensics. In the author's opinion, what are the main advantages of this proposed method compared with adversarial attack?
- Is the metric ACC in the experiment based on the threshold predefined by the authors or directly calculated by the detectors?
- Based on my understanding, image inpainting algorithms are designed to fill in the missing parts of an image.  Once this step is completed, the task of the inpainting algorithm is considered finished.  I am not at all concerned with whether someone might use detection algorithms to identify inpainted areas, nor do I care about the results of such detection.  As long as the image is restored realistically enough, the inpainting algorithm is deemed satisfactory.  I do not understand the purpose of anti-forensic capabilities, just as with a "denoising algorithm with anti-forensic functions."  This seems absurd, and I fail to see its significance;  the author appears to be pursuing innovation for its own sake.  The author's method does not surpass the current state-of-the-art inpainting algorithms, which makes it unsatisfactory.  Moreover, the author has not compared their method with inpainting algorithms from 2023 and 2024.  I will adjust my score based on the author's rebuttal.

**Suitability:**

2

---

### Official Review · Reviewer_9VRR · 2024-06-06

**Rating:** 4
**Confidence:** 4

**Summary:**

This paper proposes an image inpainting architecture with anti-forensic capabilities.The key idea is to enhance the semantic alignment between the image foreground and background using the domain adaptation method. It is experimentally verified that the anti-forensic capabilities can be significantly improved, while keeping the inpainting performance drop relatively small.

**Strengths:**

-  Introduce an end-to-end training framework for anti-forensic image inpainting, making image inpainting more secure and reliable
-  Adopt a domain adaptation method for solving the inpainting detection problem
- Performance gains are noticeable when compared with SOTA methods.

**Limitations:**

- Novelty: The model architecture almost has nothing new; all components (e.g., Domain Pattern Extractor etc.) are from existing works. Also, the only new module: RWSA module needs to be further clarified why it can improve the anti-forensic capability. The second point of contribution has not been well justified.
- It would be more interesting to add some experiments on the effect of inpainting+post-processing (low pass for example). When using inpainting to remove objects, some post-processing operations are usually employed too.
- The datasets are a bit outdated. More recent datasets could better demonstrate the superiority of the proposed method. Also, from Table 1, it seems that many latest inpaiting methods are even worse than some earlier ones, which is not quite reasonable.

**Suitability:**

2

---

### Meta-Review · Area_Chair_4kMr · 2024-07-03

**Recommendation:** Accept (Oral)
**Confidence:** 5

**Metareview:**

The paper describes an image inpainting method that (i) realistically fills the missing parts of an image and (ii) does it such that traditional forensics detection algorithm do not detect the tampering caused by inpainting.

Most modules in this work are existing, but they are assembled in an original and solid way -- Novelty is limited, yet, inspiring contributions are nicely put together. The proposed system is end-to-end.

Performance evaluation against adequate benchmarks show the gains compared to recent competitors.

Most questions raised by the reviewers were appropriately addressed in the rebuttal. Authors made several promises for the final version, including something about diffusion models as an alternative.

Overall, (confident) reviews highlight the merits of this paper, which should be accepted at the conference.

Anti-forensic inpainting is a nice target as it tackles privacy/security issues. However, it raises ethical concerns as erroneously considering that an image is genuine (i.e. pristine, authentic) may be problematic in some situations. Better motivations for coupling inpainting and forensics would be nice, as well as some higher level discussion about ethics.